# Copepod Feeding Responses to Changes in Coccolithophore Size and Carbon Content

Jordan Toullec [1,*], Alice Delegrange [1,2], Adélaïde Perruchon [1], Gwendoline Duong [3], Vincent Cornille [1], Laurent Brutier [1] and Michaël Hermoso [1]

1   Laboratoire d'Océanologie et de Géosciences—UMR 8187 LOG, CNRS, Université Littoral Côte d'Opale, F-62930 Wimereux, France
2   Institut national supérieur du professorat et de l'éducation, Académie de Lille—Hauts de France, F-59658 Villeneuve d'Ascq, France
3   Laboratoire d'Océanologie et de Géosciences—UMR 8187 LOG, CNRS, Université de Lille, F-59000 Lille, France
*   Correspondence: toullec.jordan@gmail.com

**Abstract:** Phytoplankton stoichiometry and cell size could result from both phenology and environmental change. Zooplankton graze on primary producers, and this drives both the balance of the ecosystem and the biogeochemical cycles. In this study, we performed incubations with copepods and coccolithophores including different prey sizes and particulate carbon contents by considering phytoplankton biovolume concentration instead of chlorophyll *a* level (Chl *a*) as is usually performed in such studies. The egestion of fecal pellet and ingestion rates were estimated based on a gut fluorescence method. The latter was calibrated through the relationship between prey Chl *a* level and the biovolume of the cell. Chl *a*/biovolume ratio in phytopkanton has to be considered in the copepod gut fluorescent content method. Both coccolithophore biovolume and particulate inorganic/organic carbon ratios affect the food foraging by copepods. Finally, we observed a non-linear relationship between ingestion rates and fecal pellet egestion, due to the presence of calcite inside the copepod's gut. These results illustrate that both prey size and stoichiometry need to be considered in copepod feeding dynamics, specifically regarding the process leading to the formation of fecal pellets.

**Keywords:** coccolithophore; elemental stoichiometry; copepods; gut content; ingestion rate; fecal pellet egestion; functional response

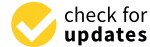



## 1. Introduction

By absorbing about 50% of carbon dioxide ($CO_2$) from the atmosphere, the Ocean plays a major role in the global carbon cycle [1]. The biological carbon pump is sustained by photosynthetic $CO_2$ fixation by phytoplankton, and by the transfer of both organic and inorganic carbon to the deep sea [2,3]. Zooplankton, as primary consumers, control the carbon transfer through excretion/respiration [4,5] producing fecal pellets that foster the export of particulate carbon flux, as observed through the analysis of sediment traps [4,6–8].

Mesozooplankton (>200 μm) prey assemblages are constituted of heterotrophic microzooplankton (flagellates, cilliates) and autotrophs such as diatoms and coccolithophores [9–11]. Coccolithophores are a key food-source group widely dispersed throughout the world's oceans [12,13]. They produce calcified structures—coccoliths, which have formed a substantial proportion of pelagic sediments since the Late Triassic period (about 200 million years ago). Fossil records show that coccolithophores were a major component of primary producers over this period, and a significant food source for zooplanktonic grazers during this period [14,15].

Both phytoplankton and zooplankton are the first to experience natural environmental shifts such as phenological changes, or anthropogenic changes induced by global warming or ocean acidification [16]. Recently, morphological changes (cell size and shape) and the

relative abundance distribution in diatom assemblage were linked to the annual phenology in the North Sea [17]. These findings could have consequences on copepod grazing [18–20]. Indeed, morphological defence of phytoplankton against grazing can be the formation of chain, cell size/shape, and biomineralization [21]. Even though copepods are well designed to break down biomineral structures and sometimes can graze on larger prey than themselves [22,23], diatom frustules limit copepod grazing [24–26] as well as microzooplankton grazing [27]. Moreover, it has been established that grazers could induce diatom silicification [28,29], proving the defensive role of the these biomineral structures. Similarly, coccolithophore build biomineral shells made of calcium carbonate (coccoliths) whose formation is influenced by environmental conditions (see reference above). As for diatom frustules, these coccoliths arranged around the cell forming the coccosphere provide mechanical protection [30], and could play a defensive role against microzooplankton grazing [31,32]. Although suggested, but never demonstrated, this calcified coccosphere could also be considered as an anti-grazing protection against copepods [33–35].

Copepods are characterised by distinct functional feeding traits (they are feeding-current feeders or ambush feeders), and as such, are interesting organisms for studying the trophodynamics towards phytoplankton [36]. Classically, ingestion rates increase with food availability and follow Ivlev's model curves [37]. This relationship is formalised by an optimal foraging theory [38,39]. The modification of copepods feeding behaviour potentially has consequences for the functioning of ecosystems, such as "trophic cascades" with consequences on biogeochemical cycles [33,40–42]. In the context of global warming and ocean acidification, a species-specific difference in coccolithophore response is expected [43] on both cell size and calcification. In this study, the modification of calcite content and cell size on copepod ingestion was explored. As a result of experimental incubations, the prediction of an optimal foraging model (Ivlev's model) was tested through direct observations of copepods' functional responses with different coccolithophore species, characterized by different calcite contents and sizes. Moreover food type and availability affect fecal pellet production rates, pellet volumes, and sinking rate, regarding compactness and mineral ballasting [44,45] (Table 1). We hypothesise that both calcite content and prey volume affect copepod functional responses and by this way, the fecal pellet egestion.

**Table 1.** Parameters indicating the initial conditions during the experimental incubations (mean $\pm$ SD, N = 3).

| Experiment | Prey Type per Incubation Batch | Cell Diameter | Chlorophyll *a* per Cell | Organic Carbon per Cell | Inorganic Carbon per Cell | Organic Nitrogen per Cell | TPC/N | POC/N | PIC/POC |
|---|---|---|---|---|---|---|---|---|---|
| | | [μm] | [pg Chl *a* cell$^{-1}$] | [pg POC cell$^{-1}$] | [pg PIC cell$^{-1}$] | [pg PON cell$^{-1}$] | [mol:mol] | [mol:mol] | [mol:mol] |
| 1 | *G. oceanica* | 6.7 $\pm$ 0.9 | 0.16 $\pm$ 0.05 | 13 | 50.7 | 4.4 [<dl] | 17.0 [ns] | 3.5 [ns] | 3.9 |
| | *G. oceanica + Tisochrysis* sp. | 6.3 $\pm$ 0.9 [a] | 0.40 $\pm$ 0.05 [a] | 25.8 | 17 | 2.6 [<dl] | 19.3 [ns] | 11.6 [ns] | 0.7 |
| | *Tisochrysis* sp. | 6.1 $\pm$ 0.6 | 0.6 $\pm$ 0.1 | 21.4 | 0 | 1.3 [<dl] | 19 [ns] | 19.0 [ns] | 0 |
| 2 | *G. oceanica* | 6.3 $\pm$ 0.9 | 0.17 $\pm$ 0.05 | 21.5 $\pm$ 1.3 | 6.7 $\pm$ 1.4 | 2.5 $\pm$ 0.1 | 13.3 $\pm$ 0.8 | 10.1 $\pm$ 0.3 | 0.3 $\pm$ 0.1 |
| 3 | *E. huxleyi* | 4.5 $\pm$ 0.5 | 0.10 $\pm$ 0.01 | 7.7 $\pm$ 0.3 | 5.4 $\pm$ 3.2 | 1.3 $\pm$ 0.2 [<dl] | 12.1 $\pm$ 4.3 [ns] | 6.9 $\pm$ 0.8 [ns] | 0.7 $\pm$ 0.4 [ns] |
| 4 | *C. braarudii* | 17 $\pm$ 2 | 3.5 $\pm$ 0.5 | 110 $\pm$ 7 | 183 $\pm$ 3 | 13.4 $\pm$ 0.8 | 25.6 $\pm$ 1.2 | 9.7 $\pm$ 0.7 | 1.7 $\pm$ 0.1 |
| 5 | *E. huxleyi* | 5.13 $\pm$ 0.03 | 0.12 $\pm$ 0.01 | 20 $\pm$ 1 | 2.6 $\pm$ 1.2 | 1.7 $\pm$ 0.1 | 15.8 $\pm$ 1.3 | 13.9 $\pm$ 0.3 | 0.13 $\pm$ 0.07 |

[a] Corresponding to the mix of both cell species. [<dl] Under the detection limit. [ns] Statistically non-significant.

## 2. Methods and Materials

### 2.1. Phytoplankton Cultures

For the laboratory experiment setup, three species of calcifying Haptophyceae were used: *Emiliania huxleyi* (strain RCC 1256); *Coccolithus braarudii* (strain RCC 1200); and *Gephyrocapsa oceanica* (strain RCC 1314). They were grown in polycarbonate flasks in 100–400 mL of K/2 + Si media at 15 °C and under a 12:12 h day:night photoperiod (100–150 μE m$^{-2}$ s$^{-1}$). The culture media were prepared with 0.2 μm filtered seawater (FSW) from the English Channel (33–34 PSU) [46]. The culture media pH was adjusted to 8.2 (total scale) by the addition of NaOH. The cells were maintained in an exponential growth phase by renewing the media every week. In parallel, non-calcifying Haptophyceae species were also cultured, *Tisochrysis* sp. (strain RCC 1350), grown inside a 2 L Erlenmeyer flask with a K/2 + Si medium at 15 °C and under a 12:12 h day:night photoperiod (100–150 μE m$^{-2}$ s$^{-1}$). These cultures were directly used after dilution with 1 μm of FSW buffered at pH 8.2 for the copepod incubation experiments (Table 1).

#### 2.1.1. Cell Count and Size Measures

Cell numeration and sizing were done using a Beckman Coulter Counter Multisizer 4E apparatus fitted with a 70 μm aperture tube. Sampled cell suspensions were diluted with an isotonic (ISOTON II) solution before being analysed. Cell sizes (cell diameter in μm) were determined by the Gaussian distribution of dominant particles present inside the culture samples (containing phytoplankton) (Table 1).

#### 2.1.2. Cell Chlorophyll *a* (Chl *a*) Content

Amounts of 100 mL of pre-diluted phytoplankton culture were filtered onto precombusted (4 h at 450 °C) glass fibre filters (Whatman GF/F) and conserved at −20 °C prior to pigment extractions. The filters were then ground overnight in 6 mL of acetone (90%) for chlorophyllian pigment extraction (Chl *a* and phaeopigments) in the dark at 4 °C. Fluorescence of the extract was measured before and after acidification with 10% HCl using a fluorometer (Turner design Trilogy). Results are expressed in pg Chl *a* $_{eq}$ cell$^{-1}$ (Table 1).

#### 2.1.3. Particulate Inorganic Carbone (PIC), Particulate Organic Carbon (POC), and Particulate Organic Nitrogen (PON)

Before each incubation, 100 mL phytoplankton culture suspensions (with known cell concentration) were filtered onto pre-combusted (4 h at 450 °C) glass fibre filters (Whatman GF/F). All the filters were then rinsed with 10 mL of FSW. Due to the large number of samples, the filters were not triplicated. The filters were placed inside aluminium foil, dried at 55 °C for 24 h, and analysed for elemental C and N using a Thermo Fisher Flash 2000 elemental analyser [47]. Two batches of glass filters were filtered for each sample, one batch with an acid treatment (providing the POC content) and the other without an acid treatment (providing the PIC + POC content), namely the total particulate carbon content, (TPC). PIC was obtained by subtracting POC from the TPC. The results are expressed in mass per cell (pg cell$^{-1}$), for inorganic carbon, organic carbon, and organic nitrogen (Table 1).

### 2.2. Copepod Sampling

For the laboratory experiments, two calanoid copepod species (*Temora longicornis* and *Acartia clausi*) were selected due to their abundance in the Eastern English Channel (EEC). Their presence generally matches phytoplankton spring blooms in the coastal areas of the EEC [48]. Each species also exhibits different functional traits [49] regarding their feeding strategies: *A. clausi* (1.1 mm total length) is an omnivorous feeding-current feeder with a clear tendency to herbivory; and *T. longicornis* (1.2 mm total length) is described as both a feeding-current feeder and cruise feeder [49,50].

The copepods were collected from February to May 2021 close to the French coast of the EEC (50°44′27.5 N: 1°34′32.4 E) during cruises on-board the N/O Sepia II (INSU-CNRS) with a WP2 plankton net (200 μm mesh size) fitted with a 2 L filtering cod-end

during horizontal net tows (speed < 1 m s$^{-1}$ for less than 10 min) at 1–3 m depth. After each plankton haul, zooplankton samples were immediately diluted in 20 L of surface seawater, then stored in the dark in a cool box and brought back within a few hours to the laboratory. To initiate the rearing phase, a ratio of 1 male per 5 females for calanoid copepods is required [51,52], and this was ensured by selecting about 250 adults of each species under a dissecting microscope. The copepods were placed in polycarbonate beakers of varying volume (from 3 to 7 L according to the number of individuals) containing 1 μm FSW. The copepods were kept at 15 °C, at a salinity of 33–34 PSU and under a 12:12 h day:night photoperiod. They were fed daily under replete food condition. The food supplied consisted of a mixture of microalgae *Rhodomonas salina* (RCC 1507), *Thalassiosira weissflogii* (RCC 1714), *Tisochrysis* sp. (RCC 1350), *Tetraselmis suesica* (RCC 1975), and *Emiliania hyxleyi* (RCC 1256), grown inside a 2 L Erlenmeyer flask with K/2 + Si medium at 15 °C and under a 12:12 h day:night photoperiod (100–150 μE m$^{-2}$ s$^{-1}$). The media were prepared with autoclaved 1 μm FSW from the EEC. The algal concentrations inside the beakers were from 10$^3$ to 10$^4$ cell mL$^{-1}$ [51–53] in order to avoid predation of calanoid copepods on younger stages [54,55]. Seawater was renewed every two days and air was supplied via small bubbles in each rearing beaker.

*2.3. Experimental Setup*

A total of eleven separate incubations of copepods (each conditions triplicated) were conducted, spread over five assays that allowed the integration of variable predator/prey size ratios and concentration ratios. Phytoplankton cell diameter ranged from 4.5 to 17 μm (Table 1) and concentrations from $1.6 \pm 0.2 \times 10^3$ cell mL$^{-1}$ to $58 \pm 2 \times 10^3$ cell mL$^{-1}$ (Table 2). The corresponding initial food concentrations ranged from $0.49 \pm 0.06$ to $10.1 \pm 2.2$ μg Chl *a* L$^{-1}$ and the total cell volume ranged from $0.39 \pm 0.03$ to $5.55 \pm 0.25$ mm$^3$ L$^{-1}$ considering the cell concentrations and their respective cell biovolume, assuming spherical cells (Table 2).

2.3.1. Copepod Selection

For each incubation, adults and copepodite 5 stage were selected corresponding to a mean length of $1097 \pm 108$ μm (N = 296) and $1216 \pm 135$ (N = 369) for *Acartia clausi* and *Temora longicornis*, respectively. In order to obtain a significant grazing signal index, copepod abundance inside bottles was high relative to calanoid copepod abundances commonly measured during phytoplankton blooms in the North Atlantic Ocean (typically 4 ind L$^{-1}$ for calanoid copepods such as *T. longicornis, A. clausi* [56]). However, the chosen experimental copepod abundance was comparable to abundances observed in the EEC (up to 11 ind L$^{-1}$, see Table 2) [57]. These high abundances remained also comparable to values used in most experimental studies ranging from 8 to >15 ind L$^{-1}$ [52,58–60].

2.3.2. Incubation

Twenty-four hours prior to the start of the experiments, 100 reared copepods were isolated in 3 L beakers containing 1 μm FSW without food. This starving phase allowed gut evacuation and maximized the feeding during the incubations. For all experiments, dead and injured individuals were first removed and only healthy-looking and living ones were individually pipetted into a 2350 mL polycarbonate bottle containing prey assemblages. Then, to avoid air bubble introduction the bottles were filled without headspace with FSW adjusted to pH 8.2, and then placed on a rolling table at 3 rpm to allow prey homogenization. Incubation was carried out at 15 °C under a photoperiod regime (12:12 h) for 24 h.

**Table 2.** Initial grazing experiment incubation setup (mean $\pm$ SD, N = 3).

| Experiment | Food | Copepod Species | Copepod per Incubation | Replicat | Cell Concentration | Chlorophyll *a* Concentration | Particulate Organic Carbon Concentration | Particulate Inorganic Carbon Concentration | Particulate Organic Nitrogen Concentration |
|---|---|---|---|---|---|---|---|---|---|
| | | | [ind L$^{-1}$] | [N] | [10$^3$ cell mL$^{-1}$] | [µg Chl *a* L$^{-1}$] | [µg POC L$^{-1}$] | [µg PIC L$^{-1}$] | [µg PON L$^{-1}$] |
| 1 | *G. oceanica* | *T. longicornis* | 13–14 | 3 | 2.9 $\pm$ 0.2 | 0.5 $\pm$ 0.1 | 37 | 146 | 13 [<dl] |
| | *G. oceanica + Tisochrysis* sp. | | 13–15 | 3 | 3.5 $\pm$ 0.3 | 1.4 $\pm$ 0 | 91 | 60 | 9 [<dl] |
| | *Tisochrysis* sp. | | 12–14 | 3 | 3.3 $\pm$ 0.3 | 2.0 $\pm$ 0.2 | 72 | 0 | 4 [<dl] |
| 2 | *G. oceanica* | *A. clausi* | 11 | 3 | 19.2 $\pm$ 0.1 | 3.2 $\pm$ 0.6 | 441 $\pm$ 25 | 128 $\pm$ 27 | 47 $\pm$ 1 |
| | | *T. longicornis* | 11 | 3 | | | | | |
| 3 | *E. huxleyi* (low concentration) | *A. clausi* | 11–14 | 3 | 13.7 $\pm$ 0.8 | 1.4 $\pm$ 0 | 104 $\pm$ 4 | 73 $\pm$ 43 | 18 $\pm$ 3 [<dl] |
| | | *T. longicornis* | 11–13 | 3 | | | | | |
| 4 | *C. braarudii* | *A. clausi* | 14–18 | 3 | 2.2 $\pm$ 0.9 | 7.6 $\pm$ 0.2 | 244 $\pm$ 16 | 405 $\pm$ 7 | 30 $\pm$ 2 |
| | | *T. longicornis* | 11–17 | 3 | | | | | |
| 5 | *E. huxleyi* (high concentration) | *A. clausi* | 16–19 | 3 | 57.9 $\pm$ 0.2 | 7.3 $\pm$ 0.4 | 1157 $\pm$ 57 | 151 $\pm$ 69 | 97 $\pm$ 7 |
| | | *T. longicornis* | 11 | 3 | | | | | |

[<dl] Under the detection limit.

*2.4. Ingestion/Egestion Estimation*

After each incubation, the copepods were carefully retrieved from each bottle by sieving the seawater through an immersed 200 μm mesh. The copepods were placed in 2 mL cryotubes (one per bottle) and then flash frozen in liquid nitrogen and kept at –20 °C until further analysis. Copepod size measurements were performed (as much as possible not withstanding obscurity) under a dissecting microscope (ZEISS Axio Zoom V16), before pigment extraction for gut content quantification (see below). Fecal pellets were recovered after each incubation by filtering the remaining seawater of each bottle onto a 40 μm mesh sieve. Fecal pellets retained on the mesh sieve were resuspended in FSW in a plankton counting chamber (Dolfuss cuvette, 6 mL volume).

2.4.1. Copepod Gut Pigment Content

For gut content analyses, copepods were individually sorted from freshly thawed samples under a cool light stereomicroscope. Individuals were rinsed with 0.2 μm FSW to eliminate phytoplankton cells with aggregates stuck to feeding appendages and were then transferred into 4 mL acetone (90%). Individuals (N = 19 to 42 copepods per extraction) were ground and chlorophyllian pigments (Chl *a* and phaeopigments) were extracted in the dark at 4 °C overnight. Fluorescence of the extract was measured before and after acidification with 10% HCl using a fluorometer (Turner design Trilogy). Copepod gut content was obtained by both Chl *a* and phaeopigment concentrations and values were not corrected for pigment degradation on the recommendation of Durbin and Campbell [61]. Ingestion rates ($I$, ng Chl $a$ $_{eq}$ ind$^{-1}$ d$^{-1}$) were derived from gut total pigment content ($G_{cop}$, ng Chl $a$ $_{eq}$ ind$^{-1}$) using Equation (1):

$$I = 60 \times G_{cop} \times k \qquad (1)$$

where $k$ is the gut evacuation rate (h$^{-1}$), calculated following the model of Dam and Peterson [62], which accounts for the temperature of incubation, and the specie-dependant allometric constant. In the present study, we carried out our calculations with $k = 0.028$, which corresponds to the allometric constant of evacuation of calanoids at 15 °C ($k = 0.0117 + 0.001794 \times$ T).

2.4.2. Copepod Gut Volume Conversion

Phytoplankton species used during the grazing experimental setup did not have the same biovolume and Chl *a* content (see Table 1). In order to compare every gut content for each experiment, we converted the equivalent pigment gut content (ng Chl $a$ $_{eq}$ ind$^{-1}$) into volume equivalent gut content (μm$^3$ $_{eq}$ ind$^{-1}$). A calibration of Chl *a* level (pg Chl a cell$^{-1}$) over cell biovolume (μm$^3$) for each phytoplankton species was used. Gut ingestion was then expressed as its prey biovolume equivalent (10$^6$ μm$^3$ $_{eq}$ ind$^{-1}$ d$^{-1}$).

2.4.3. Ivlev's Model

The copepod ingestion functional response toward food availability was calculated by following Ivlev's model [38,39]—Equation (2). This model considers the optimal food foraged by copepods (and more widely by all planktonic active filter feeders), recently described as a *Type II functional response* [63,64].

$$I_{Ivlev} = I_{max} \times \left( 1 - e^{(-\alpha \times C_{food})} \right) \qquad (2)$$

where $I_{max}$ is the maximum ingestion rate index obtained; $\alpha$ the rate at which saturation is achieved with increasing food levels (slope of the linear regression); $I_{Ivlev}$ is the modelized ingestion rate; and $C_{food}$ is the corresponding food concentration (μg POC L$^{-1}$, μg Chl $a$ L$^{-1}$ or mm$^3$ L$^{-1}$).

### 2.4.4. Fecal Pellet Production and Size

Fecal pellet production (FP $ind^{-1}$ $d^{-1}$) was estimated after each experiment by counting the fecal pellets recovered after incubation. For each incubation, between 10 and 186 pellets were measured (length and width in µm) with 5 µm accuracy. Fecal pellets are considered as cylindrical with two half spheres, and volumes were calculated according to Equation (3) [65]:

$$V_{PF} = \pi \times d^2 \times \left( \frac{L}{4} + \frac{d}{6} \right) \tag{3}$$

where $d$ is the pellet diameter (µm), and $L$ is the length of cylindric part of the pellet. Volumes were then converted into equivalent spherical diameter (ESD, mm), according to Equation (4):

$$ESD = \sqrt[3]{\frac{6 \times V}{\pi}} \tag{4}$$

### 2.5. Statistical Analyses

Results are expressed in mean $\pm$ standard deviation (SD). When data distribution matched the parametric assumption of normality (tested with a Shapiro–Wilk test, $p < 0.05$), correlation between two variables was analysed using a Pearson correlation test. Otherwise, a Spearman rank correlation test was performed. The statistical effect of the different experimental conditions was tested with a one-way ANOVA, followed by a pairwise Tukey's *post hoc* comparison test. In case of non-normal distribution, multicomparaisons were performed using the Kruslal–Wallis test following Nemeyi *post hoc* test. All the statistical analysis was performed using R software (V 4.1.1).

## 3. Results

### 3.1. Coccolithophore Stochiometry

Cellular particulate organic carbon (pg POC $cell^{-1}$), nitrogen (pg PON $cell^{-1}$), and inorganic carbon (pg PIC $cell^{-1}$) increase with coccolithophore diameter (Table 1). Cellular content and stoichiometric ratios for each experiment and coccolithophore species are also presented in Table 1. The cells are considered as spheres, whose biovolumes varied from 38 to 83 $µm^3$ for *E. huxleyi* (RCC 1256), from 133 to 143 $µm^3$ for *G. oceanica* (RCC 1314), and from 2296 to 2487 $µm^3$ for *C. braarudii* (RCC 1200).

### 3.2. Copepod Ingestion

For experiment 1, a mixture of coccolithophores (*G. oceanica*, RCC 1314) and non-coccolithophores (*Tisochrysis* sp. RCC 1350) was incubated with the copepods (*T. longicornis*). These two haptophyte species have similar sizes (6.7 µm and 6.1 µm of diameter for *G. oceanica* and *Tisochrysis* sp., respectively). They were mixed to obtain three batches: 100% *G. oceanica*, 50% *G. oceanica* + 50% *Tisochrsis* sp., and 100% *Tisochrysis* sp. with approximately 3000 cell $mL^{-1}$ in total (see Table 2). The cell density (cell $mL^{-1}$) and total cell volume ($mm^3$ $L^{-1}$) were non-significantly different between the three different conditions (Figure 1A) with an average of $3255 \pm 292$ cell $mL^{-1}$ and $0.44 \pm 0.04$ $mm^3$ $L^{-1}$, respectively. Concerning the Chl *a* concentration, *G. oceanica* incubation contained $0.49 \pm 0.06$ µg Chl *a* $L^{-1}$, mix of *G. oceanica* and *Tisochrysis* sp. contained $1.40 \pm 0.01$ µg Chl *a* $L^{-1}$, and *Tisochrysis* sp. contained $2.04 \pm 0.17$ µg Chl *a* $L^{-1}$ (Figure 1C). Particulate matter composition (µg POC, PIC, and PON $L^{-1}$) was achieved within the three different conditions (Figure 1D) and is presented in Table 2.

The resulting ingestion rates varied from 0 (under detection limit) to $13.1 \pm 1.4$ ng Chl *a* $_{eq}$ $ind^{-1}$ $d^{-1}$ with the higher values encountered in the 100% *Tisochryisis* sp. condition. Volume equivalent ingestion rates (Table 1, Figure 2B) varied from 0 (under the detection limit) to higher values for incubation with *Tisochrysis* sp ($2.5 \pm 0.3 \times 10^6$ $µm^3$ $_{eq}$ $ind^{-1}$ $d^{-1}$). The egestion rates were not significantly different between conditions, with averaged values of $26 \pm 7$ fecal pellets in $d^{-1}$, and mean pellet volumes ranging significantly from $0.3 \pm 0$ with *G. oceanica*, to $1.6 \pm 0.6$ $mm^3$ with *Tisochrysis* sp. (Figure 2D).

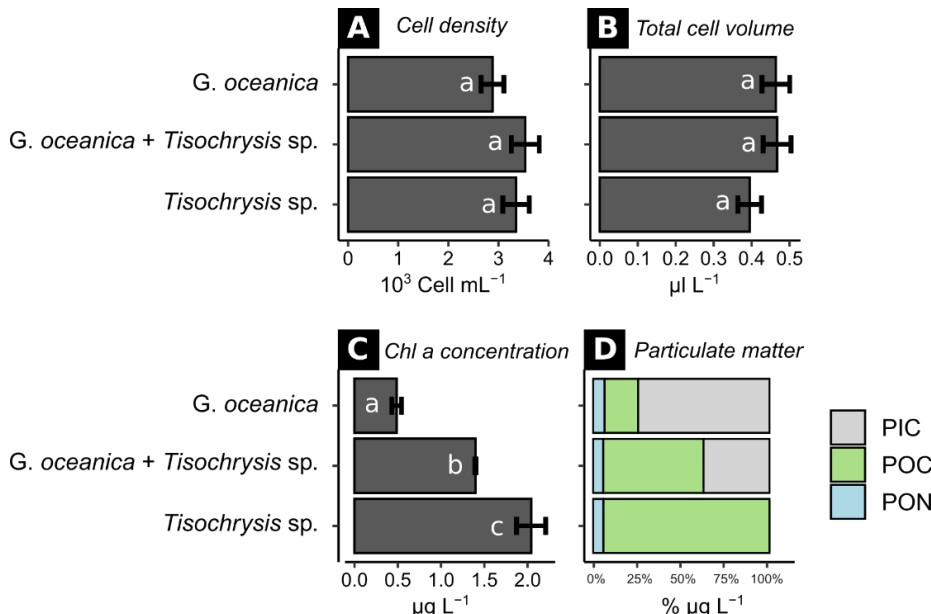

**Figure 1.** Experiment 1 incubations with *Temora longicornis*. (**A**) Initial cell density for each condition. (**B**) Initial cell volume for each condition for each condition. (**C**) Initial Chl *a* concentration for each condition. (**D**) Initial particulate matter quality for each condition. Groups a, b, and c correspond to statistical groups, according to one-way ANOVA with significant threshold α = 5% (*p*-value < 0.05).

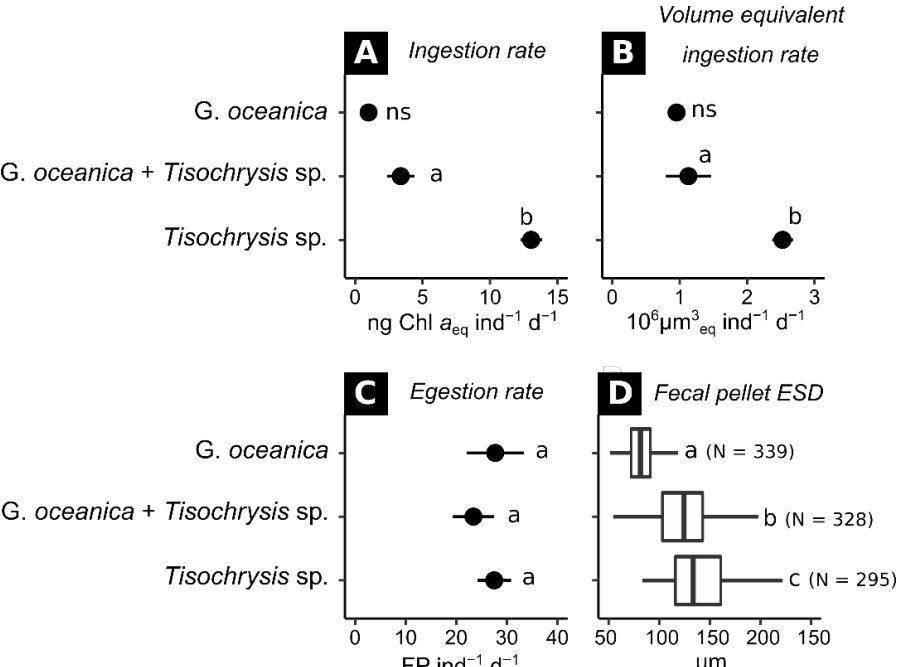

**Figure 2.** Details of Experiment 1 incubations with *Temora longicornis*. (**A**) Ingestion rate for each condition. (**B**) Volume equivalent ingestion rate for each condition. (**C**) Egestion rate for each condition. (**D**) Fecal pellet equivalent spherical diameter (ESD). Group a, b, and c correspond to statistical group, according to one-way ANOVA with significant threshold α = 5% (*p*-value < 0.05). ns = non-significant.

After the incubations, the recovered fecal pellets had both significantly different sizes (Figures 2D and 3) and different opacity: when copepods were fed with 100% *G. oceanica*, fecal pellets were opaque and thick whereas they were light green with *Tisochrysis* sp. Fecal

pellets had an intermediate aspect where the copepods were fed with a mix of both species (Figure 3).

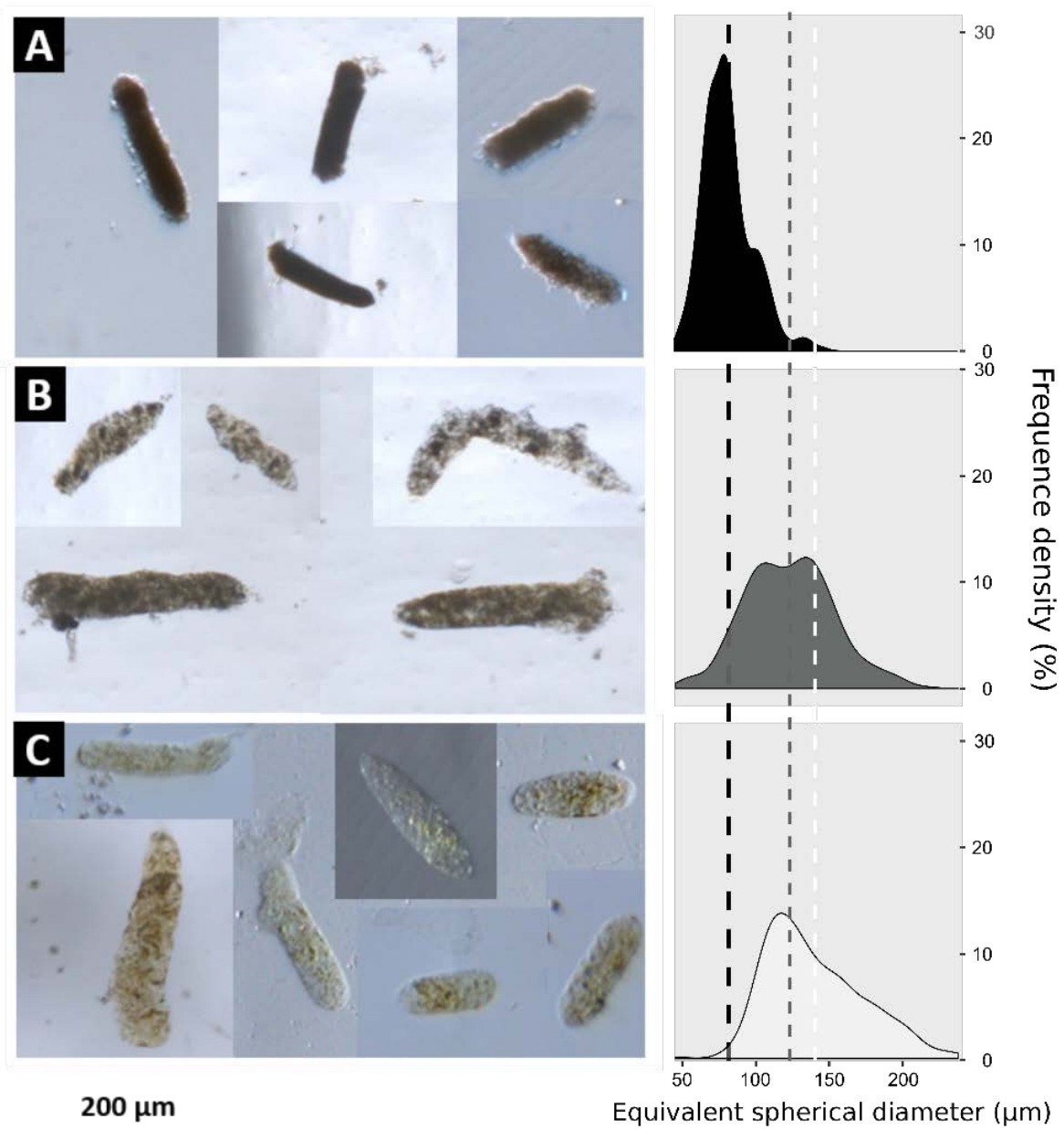

**Figure 3.** Picture of the recovered *Temora longicornis* fecal pellets of Experiment 1, after different conditions: (**A**) grazing experiment with 100% *G. oceanica*; (**B**) Grazing experiment with a mixture of 50% *G. oceanica* + 50% *Tisochrysis* sp.; and (**C**) Grazing experiment with *Tisochrysis* sp. The scale bar is congruent with figure (**A–C**). The vertical black dashed line corresponds to the mean fecal pellet diameters (µm) recovered after grazing experiment with 100% *G. oceanica* (**A**); the vertical grey dashed line corresponds to the mean fecal pellet diameters (µm) recovered after grazing experiment with a mixture of 50% *G. oceanica* + 50% *Tisochrysis* sp (**B**); the vertical white dashed line corresponds to the mean fecal pellet diameters (µm) recovered after grazing experiment with *Tisochrysis* sp (**C**).

The following figures (Figures 4–8) and results consider all the experiments.

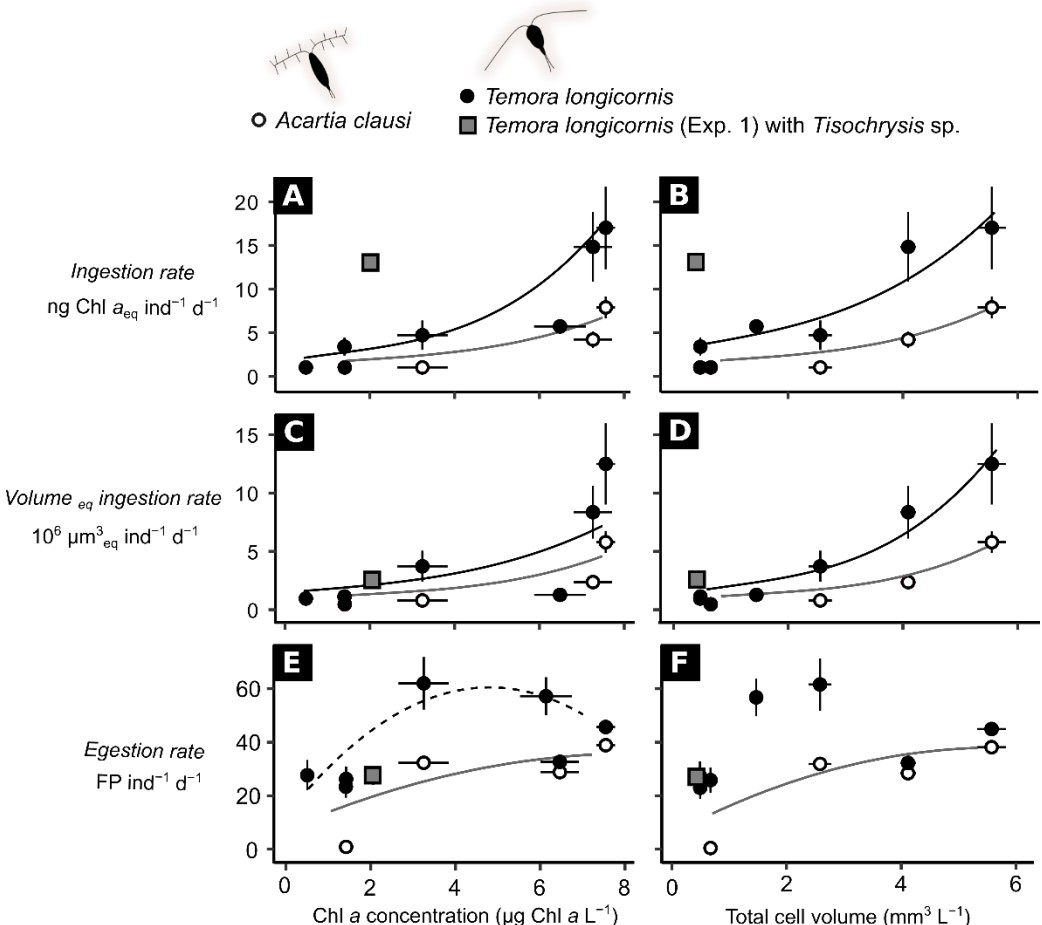

**Figure 4.** Copepod functional responses over initial Chl *a* concentration (μg Chl *a* L$^{-1}$) are shown on the left, and over initial total cell volume (mm$^3$ L$^{-1}$) are shown on the right. (**A,B**) depict the pigment ingestion rate (ng Chl *a* $_{eq}$ ind$^{-1}$ d$^{-1}$); (**C,D**) depict the volume equivalent ingestion rate (10$^6$ μm$^3$ $_{eq}$ ind$^{-1}$ d$^{-1}$); and E and F depict the fecal pellet egestion rate (FP ind$^{-1}$ d$^{-1}$). Black circular dots correspond to incubation with *Temora longicornis*. White circular dots correspond to incubation with *Acartia clausi*. Grey square dots correspond to incubation with *Temora longicornis* and *Tisochrysis* sp. cell (monospecific and mixed with *G. oceanica*). Solid lines represent exponential fit of pigment/volume equivalent ingestion rate over Chl *a* concentration and total cell volume (**A–D**). In (**E,F**), solid lines correspond to quadratic fit of fecal pellet egestion over Chl *a* concentration and total cell volume. All equations and statistics are displayed in Table 3. In all graphs, *p*-values < 0.001 are displayed by solid lines, however, *p*-values < 0.05 are displayed by dashed lines (see statistical test in the Methods Section).

The functional responses to prey concentration varied significantly between those copepod species with an average lower ingestion and fecal pellet egestion by *A. clausi* compared to those with *T. longicornis* (Figure 4). Including all experiments, the Chl *a* concentration ranged from 0.49 ± 0.06 to 7.6 ± 0.2 μg Chl *a* L$^{-1}$ (Figure 4A,C,E). The total cell volume ranged from 0.39 ± 0.03 to 5.5 ± 0.3 mm$^3$ L$^{-1}$ (Figure 4B,D,F). In parallel, the ingestion rate values increased from 0 to 9.9 ng Chl *a* $_{eq}$ ind$^{-1}$ d$^{-1}$ for *Acartia clausi* and from 0 to 23.1 ng Chl *a* $_{eq}$ ind$^{-1}$ d$^{-1}$ for *Temora longicornis* (Figure 4A,B). The volume equivalent ingestion rate values ranged from 0.46 to 7.3 × 10$^6$ μm$^3$ $_{eq}$ ind$^{-1}$ d$^{-1}$ for *Acartia clausi* and from 0.9 to 17 × 10$^6$ μm$^3$ $_{eq}$ ind$^{-1}$ d$^{-1}$ for *Temora longicornis* (Figure 4C,D). The fecal pellet egestion rate ranged from 4 to 41 FP ind$^{-1}$ d$^{-1}$ for *Acartia clausi* and from 19 to 76 FP ind$^{-1}$ d$^{-1}$ for *Temora longicornis* (Figure 4E,F). All fits and statistical parameters are displayed in Table 3.

**Table 3.** Regression and statistical parameters within functional relation between food level index and ingestion rate index.

| Food Level Index | Grazing Rate Index | Copepod | Statistics | |
|---|---|---|---|---|
| | | *T. longicornis* | **Equation** | $R^2$ |
| *Ingestion rate index* | | | *Exponential fit* | |
| Chl *a* concentration (µg Chl *a* L$^{-1}$) | Pigment ingestion rate (ng Chl *a* $_{eq}$ ind$^{-1}$ d$^{-1}$) | | $y = 2.2 \times 10^{-5} e^{(1.46x)} + 4.49$ | 0.64 *** |
| | Volume equivalent ingestion rate ($10^6$ µm$^3$ $_{eq}$ ind$^{-1}$ d$^{-1}$) | | $y = 7.4 \times 10^{-7} e^{(2.19x)} + 1.6$ | 0.67 *** |
| Total cell volume (mm$^3$ L$^{-1}$) | Pigment ingestion rate (ng Chl *a* $_{eq}$ ind$^{-1}$ d$^{-1}$) | | $y = 3.1 e^{(0.31x)} + 0.82$ | 0.65 *** |
| | Volume equivalent ingestion rate ($10^6$ µm$^3$ $_{eq}$ ind$^{-1}$ d$^{-1}$) | | $y = 3.02 e^{(0.29x)} - 2.4$ | 0.85 *** |
| *Egestion rate index* | | | *Quadratic fit* | |
| Chl a concentration (µg Chl *a* L$^{-1}$) | Egestion rate (FP ind$^{-1}$ d$^{-1}$) | | $y = -2.53x^2 + 24.34x$ | 0.47 * |
| Total cell volume (mm$^3$ L$^{-1}$) | | | $y = -6.05x^2 + 39.3x$ | 0.38 $^{ns}$ |
| | | *A. Clausi* | | |
| *Ingestion rate index* | | | *Exponential fit* | |
| Chl *a* concentration (µg Chl *a* L$^{-1}$) | Pigment ingestion rate (ng Chl *a* $_{eq}$ ind$^{-1}$ d$^{-1}$) | | $y = 3.9 \times 10^{-8} e^{(2.52x)} + 0.99$ | 0.81 *** |
| | Volume equivalent ingestion rate ($10^6$ µm$^3$ $_{eq}$ ind$^{-1}$ d$^{-1}$) | | $y = 1.5 \times 10^{-9} e^{(2.9x)} + 0.55$ | 0.78 ** |
| Total cell volume (mm$^3$ L$^{-1}$) | Pigment ingestion rate (ng Chl *a* $_{eq}$ ind$^{-1}$ d$^{-1}$) | | $y = 0.43 e^{(0.53x)} + 0.06$ | 0.85 *** |
| | Volume equivalent ingestion rate ($10^6$ µm$^3$ $_{eq}$ ind$^{-1}$ d$^{-1}$) | | $y = 0.126 e^{(0.683x)} + 0.22$ | 0.85 *** |
| *Egestion rate index* | | | *Quadratic fit* | |
| Chl a concentration (µg Chl *a* L$^{-1}$) | Egestion rate (FP ind$^{-1}$ d$^{-1}$) | | $y = -1.3x^2 + 13.7x$ | 0.76 ** |
| Total cell volume (mm$^3$ L$^{-1}$) | | | $y = -0.84x^2 + 10.9x$ | 0.87 *** |

ns = non-significant, * *p*-value < 0.05, ** *p*-value < 0.01 and *** *p*-value < 0.001, according to Pearson correlation test.

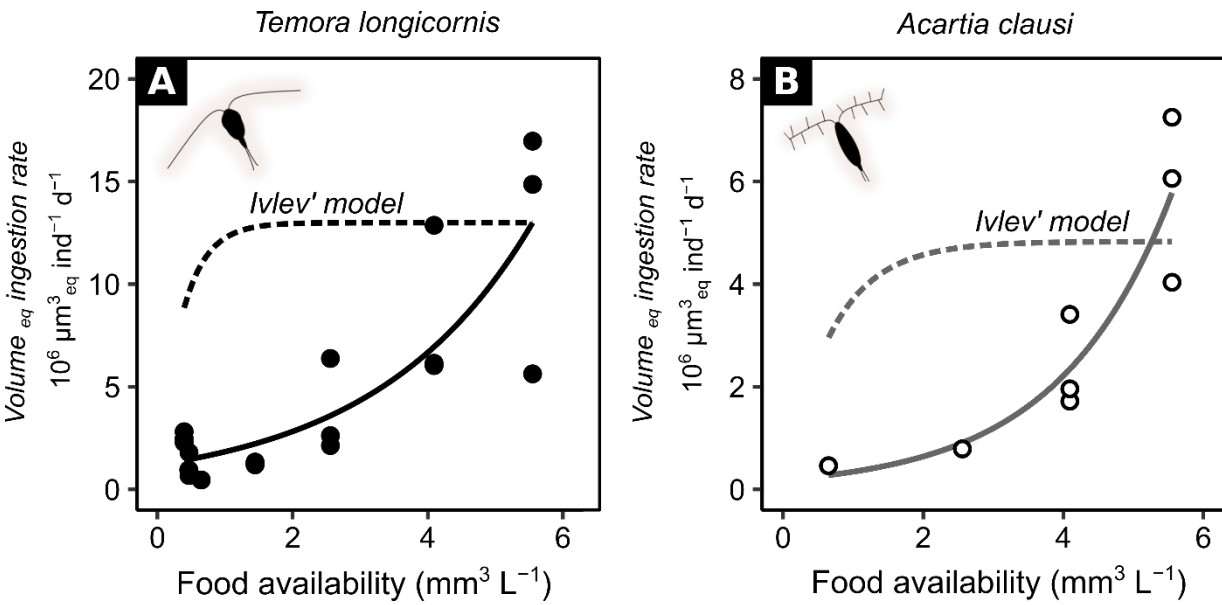

**Figure 5.** Copepod functional responses over total cell volume (mm$^3$ L$^{-1}$), for *Acartia clausi* (**A**) and *Temora longicornis* (**B**). Solid lines represent exponential fit of volume equivalent ingestion rate (10$^6$ μm$^3$ $_{eq}$ ind$^{-1}$ d$^{-1}$) over total cell volume (mm$^3$ L$^{-1}$). Dashed lines correspond to Ivlev's model considering the max ingestion rate and the increasing ingestion rate over the food level slope (Equation (2)).

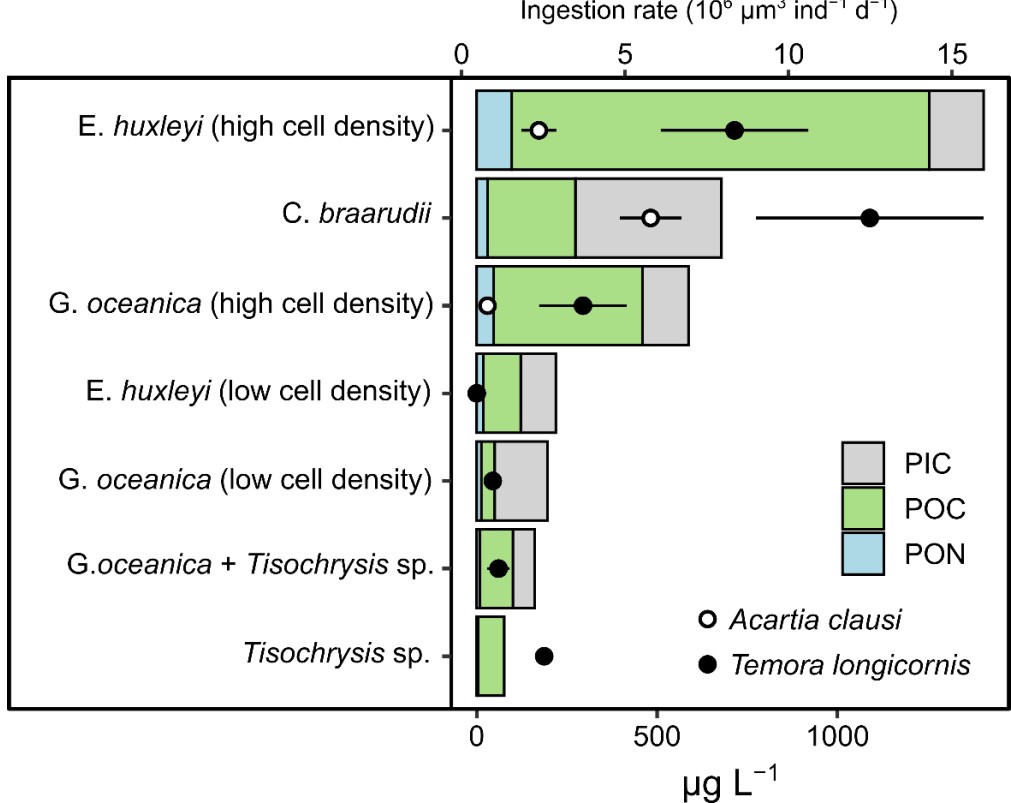

**Figure 6.** Cumulative barplot of particulate organic nitrogen (PON), particulate organic carbon (POC), and particulate inorganic carbon (PIC) in μg L$^{-1}$ for each experimental incubation (bottom axis). Depicted on the top *x*-axis: the scatterplot of the ingestion rate (10$^6$ μm$^3$ $_{eq}$ in$^{-1}$ d$^{-1}$) with *Acartia clausi* (white dots) and *Temora longicornis* (black dots).

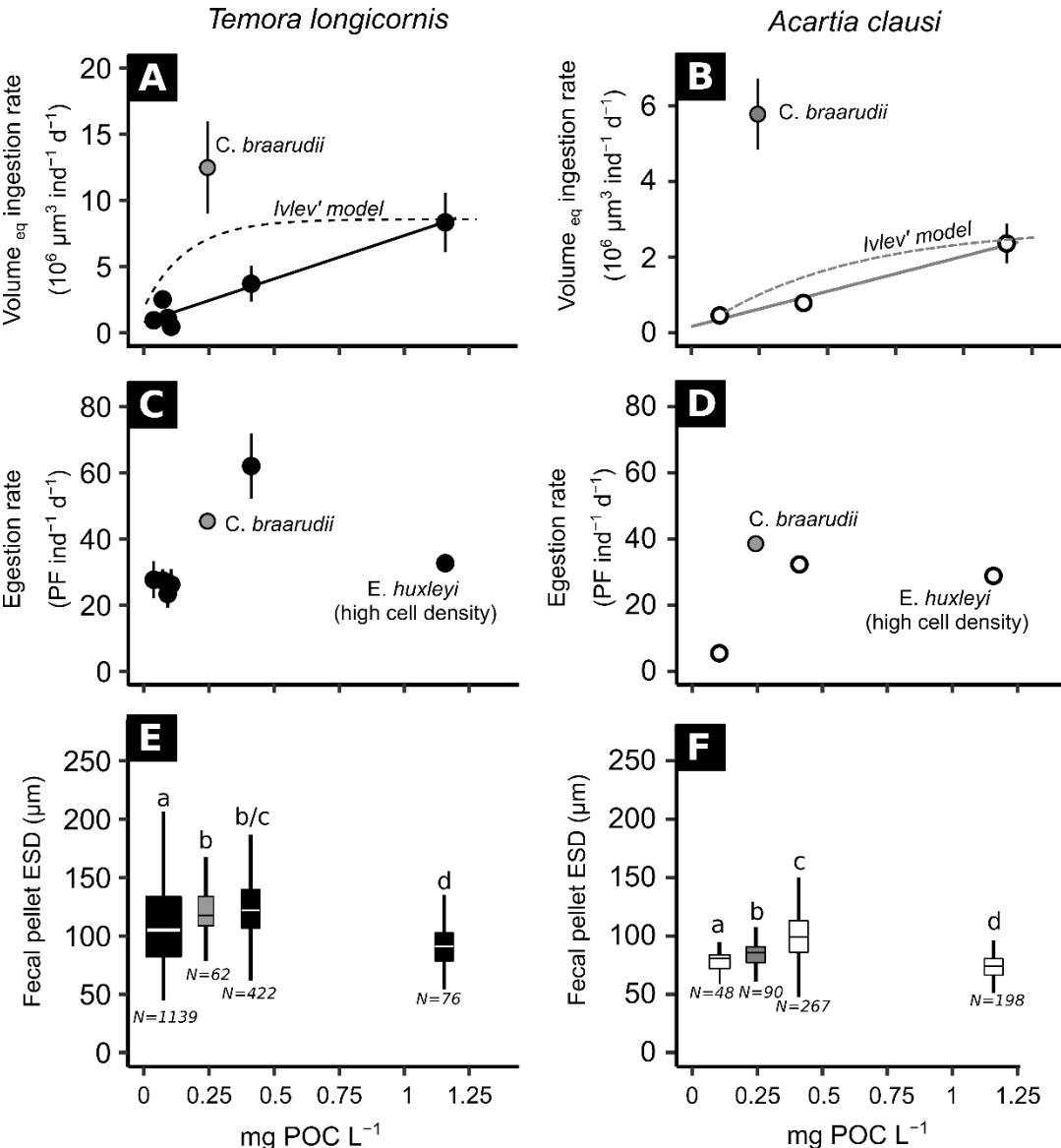

**Figure 7.** (**A**,**B**) Ingestion rate ($10^6$ $\mu m^3$ eq $ind^{-1}$ $d^{-1}$); (**C**,**D**) egestion rate (FP $ind^{-1}$ $d^{-1}$) and (**E**,**F**) fecal pellet ESD ($\mu m$) over POC concentration (mg POC $L^{-1}$) for *Temora longicornis* (**A**,**C**,**E**) and *Acartia clausi* (**B**,**D**,**F**). The grey-boxed dots correspond to Experiment 4 with *C. braarudii*. The solid lines in (**A**,**B**) represent the linear regression between POC concentration and the ingestion rate, excluding the grey circular dot (Experiment 4 with *C. braarudii*). For *T. longicornis*, Pearson $R^2$ = 0.86, N = 12, *p*-value = 0.001, for *A*. clausi, Pearson $R^2$ = 0.88, N = 12, *p*-value = 0.002 Dashed lines correspond to Ivlev's model considering the max ingestion rate and the increasing ingestion rate over the food level slope (Equation (2)). Letters a, b, c, and d (in (**E**,**F**)) correspond to the different statistical groups displayed by the Kruskall–Wallis test.

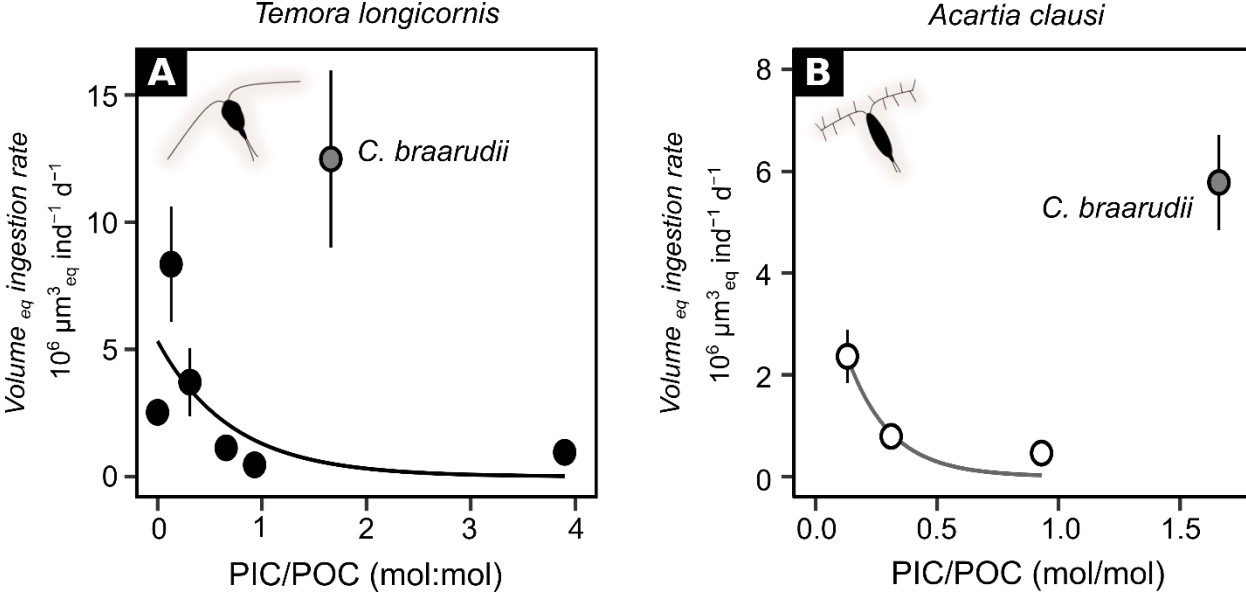

**Figure 8.** (**A**) Volume equivalent ingestion rate ($10^6$ μm$^3$ $_{eq}$ ind$^{-1}$ d$^{-1}$) over PIC/POC ratios (mol:mol) for incubation with *T. longicornis (Black dots)*; (**B**) volume equivalent ingestion rate ($10^6$ μm$^3$ $_{eq}$ ind$^{-1}$ d$^{-1}$) over PIC/POC ratios (mol:mol) for incubation with *A. clausi* (white dots). Grey dots correspond to incubation with *C. braarudii*. For incubations with *T. longicornis*, Kendall τ = −0.59, N = 18, *p*-value = 0.002, excluding the grey dot (Exp. 4 with *C. braarudii*). For the incubation with *A. clausi*, Kendall τ = −0.95, N = 9, *p*-value < 0.001, excluding the grey dot (Exp. 4 with *C. braarudii*).

The food availability varied between 0.39 ± 0.03 and 5.5 ± 0.3 mm$^3$ L$^{-1}$, and the ingestion rate varied from 0.46 to 7.3 × $10^6$ μm$^3$ ind$^{-1}$ d$^{-1}$ for *A. clausi* and from 0.9 to 17.0 × $10^6$ μm$^3$ ind$^{-1}$ d$^{-1}$ for *T. longicornis* (Figures 4 and 5). Exponential fits (solid lines) for both copepods (see Table 3) correspond to *Type III functional response* [63]. The logarithmic fits (dashed lines) were expected (Figure 5), following optimal food foraging (Ivlev's model or *Type II functional response*).

Particulate inorganic carbon (PIC) ranged from 0 (*Tisochrysis* sp. under detection limit) to 433 μg L$^{-1}$ (Figure 6). Particulate organic carbon (POC) ranged from 37 to 1157 μg L$^{-1}$ (Figure 6). Particulate organic nitrogen (PON) ranged from 4 to 97 μg L$^{-1}$ (Figure 6). During Experiment 3 (*C. braarudii*) the PIC concentration represented 55% of the total particulate pool (with 433 μg PIC L$^{-1}$ over 300 μg POC L$^{-1}$ and 55 μg PON L$^{-1}$). Except for Experiment 5, the copepod ingestion rate increased with increasing ambient particulate content (Figure 6).

When considering all the experiments—except the one with *C. braarudii* (grey dots in Figures 7 and 8)—POC concentration and equivalent volume ingestion rates are positively correlated for both copepod species (*T. longicornis*: y = 0.006x + 0.81; $R^2$ = 0.71; *p*-value < 0.001 Pearson correlation test, and with *A. clausi*: y = 0.002x + 0.17; $R^2$ = 0.77; *p*-value = 0.002 Pearson correlation test). Logarithmic fits were expected following theoretical optimal food foraging (Figures 4, 5 and 7), dashed lines: Ivlev's model or *Type II functional response*. Nevertheless we observed linear regression between POC concentration and equivalent volume ingestion rates which are described in the literature as *Type I functional response* [63]. Fecal pellet egestion ranged from 4 to 41 FP ind$^{-1}$ d$^{-1}$ for *Acartia clausi* and from 19 to 76 FP ind$^{-1}$ d$^{-1}$ for *Temora longicornis* (Figure 7 C,D). Despite a positive correlation between POC concentration and equivalent volume ingestion rates for both copepods, fecal pellet egestion was not correlated to POC concentration (Figure 7C,D). Moreover, within incubations with *T. Longicornis* and *E. Huxleyi* with high cell density (Exp. 5), despite high volume equivalent ingestion rate (8.3 ± 3.9 × $10^6$ μm$^3$ $_{eq}$ ind$^{-1}$ d$^{-1}$), fecal pellet egestion remained low (32 ± 3 FP ind$^{-1}$ d$^{-1}$). After incubation with *T. longicornis*,

the mean fecal pellet ESD ranged from 90 ± 17 to 124 ± 25 mm, whereas, after incubation with *A. clausi*, the mean fecal pellet ESD ranged from 71 ± 11 to 97 ± 20 mm.

Variations of cell calcite content were expressed as PIC/POC ratios (mol:mol) for each experiment (see Tables 1 and 2). Volume equivalent ingestion rate ($10^6$ $\mu m^3$ $_{eq}$ ind$^{-1}$ d$^{-1}$) over PIC/POC ratios (mol:mol) decreased non-linearly for both *T. longicronis* and *A. clausi* (Figure 8), when excluding experiments performed with *C. braarudii*.

## 4. Discussion

Copepod and more widely zooplankton food foraging is the main prey/predator quantifiable interaction. Within marine planktonic ecosystems, these trophic relationships have direct consequences on population dynamics for both preys and predators [66,67]. Indeed, copepod behavioural adaptations affect both ecological dynamics and biogeochemical cycles, such as primary production and sinking particles fluxes [4,68,69]. Numerical models have shown that copepods could also affect the phytoplankton prey population diversity via a top-down control [70], and also the seasonal succession of plankton communities [19].

### 4.1. Equivalent Volume Ingestion Estimation

Pigment ingestion rates based on copepod gut fluorescent content [62,71] represent a fast and easy workable way to estimate grazing. Regarding incubation times, which were equivalent in all incubations, pigment destruction inside the gut increased with gut ingestion [72], suggesting that loss of fluorescence is equivalent among the different samples from the different incubations. However, ingested preys could present significant variation in fluorescence, especially in situ; due to ingestion of non-chlorophyllian prostists (ciliates, heterotropic flagellates, nauplii) and algae (diatoms, haptophyceae) in varying proportions. In order to compare the ingestion rates derived from all the experiments (regardless of phytoplankton cell Chl *a* content and their biovolume), we used a conversion of gut content considering Chl *a* and biovolume (see Methods Section). This allowed us to get a better correlation between ingestion and total cell volume ($r^2 = 0.85$ *** for both A. *clausi* and T. *longicornis*, Table 3) than the pigment ingestion rate in accordance to Chl *a* concentration (r = 0.64 *** and 0.81 *** for T. *longicornis* and A. *clausi*, respectively; Table 3, Figure 5). Regarding these findings, we assume that the probability of prey/predator contact is more dependent on total cell volume than the number of particles (cell L$^{-1}$), or biomass (g Chl *a* L$^{-1}$ or g POC L$^{-1}$). Thus, it can be assumed that the total cell volume per litre (mm$^3$ L$^{-1}$) represents a better index of the prey-encounter rate. This suggests that, at equivalent total cell concentrations, the same ingestion rate pattern, expressed in volume equivalent Chl *a* ($\mu m^3$ $_{eq}$ ind$^{-1}$ d$^{-1}$), could be expected with large cells at low concentration as well as with small cells at high concentration. However, gut analysis neither takes into account pigment degradation inside the copepod's gut [61,72] before ingestion nor sloppy feeding (cell fragmentation without ingestion, see pictures in Jansen, 2008). Considering the very short gut passage time (less than an hour) and the relative evidence of viable cell preservation inside fecal pellets [73,74], the pigment degradation could be neglected (the same condition of sample preservation and treatment).

### 4.2. Adaptive Functional Response

Classically, the ingestion rate index based on gut content over food availability, which provides Ivlev's model curve [37], represents the optimal foraging behaviour, even regarding incubation time and pigment destruction inside the gut [72]. In this study, both prey/predator size ratios and prey stoichiometry modulate the ingestion rate index. A *Type III functional response* was obtained with both *A. clausi* and *T. longicornis* when considering food availability by total cell volumes and Chl *a* concentration (Figures 5 and 6). Indeed, ingestion rates increased exponentially according to food availability [63,64]. This functional relationship reflects a switching adaptation considering the food quality. The maximum food level reached 5 mm$^3$ L$^{-1}$, and is comparable to the maximum food availability in the literature (4 mm$^3$ L$^{-1}$) in Kiørboe et al. [64], when the copepod's ingestion saturation

occurs (e.g., *Acartia tonsa*, *Temora longicornis*, *Centropages hamatus*, and *Oithona davisae*). At high food concentration (7.6 ± 0.2 μg Chl *a* L$^{-1}$), we assume a saturation of the feeding activity. Indeed, with more than 7 μg Chl *a* L$^{-1}$, the bottles showed green coloration. In our study, we exceeded 5 mm$^3$ L$^{-1}$ at 15 °C. Thus, performing additional experiments at higher food concentrations would have had no benefit. POC, PIC, and PON quota per cell compare well to those presented in the literature, with a magnitude from 1 to 10$^2$ pgC cell$^{-1}$ and 10$^{-1}$ to 10 pgN cell$^{-1}$ [75,76]. For both *T. longicornis* and *A. clausi*, the ingestion rates increased linearly with POC concentrations (Figure 7A,B), when excluding experiments with *C. braarudii*, 17 μm diameter. This suggests a *Type I functional response* [63] corresponding to a linear increase of ingestion according to food availability. This is the most common behaviour attributed to planktonic active filter feeders (such as copepods). However, calanoid copepods (such as *Acartia* spp., *Temora* spp., *Centropages* spp., and *Calanus* spp.) commonly present *Type I* and *II* functional responses, mainly corresponding to the Ivlev model [77–80]. Ivlev's model is shaped like *Type II functional response*, which corresponds to the optimal feeding behaviour towards high-quality food availability. In this study, any relationship (either considering Chl *a* concentration, total volume, or POC concentration) fits with Ivlev's model (or *Type II functional response*) suggesting an anti-grazing propriety of the coccolithophores as a food source alone. Within the six incubations with *C. braarudii*, regarding food availability as equivalent carbon, Chl *a* or total volume, we obtained higher ingestion rates than for smaller coccolithophore, which can be explained by an intense gut accumulation of algae material because of the large cell size. Calcite cell content through PIC/POC ratio (mol:mol) for coccolithophore cells of similar sizes (Figure 8) could partially explain a non-optimal ingestion pattern observed in our experiments. These results suggest that the coccosphere (i.e., calcified exosqueletton around the cell) could be a structure protecting the coccolithophore from grazing by copepods, such as diatom frustules, as previously proposed [33,34].

### 4.3. Calcite Obstruction and Potential Dissolution Inside Copepod Guts

While the copepods ingested large coccolithophore (*C. braarudii*), we measured high ingestion rates despite low carbon concentration and low fecal pellet egestion. This observation indicates a decoupling between ingestion rate and gut passage time [80], probably due to high calcite ingestion and a decrease in gut pH resulting from calcite dissolution. This phenomenon may explain an importance paradox in ocean zooplankton mediated calcite dynamics. Indeed, considering a global oceanic alkalinity budget, there is a loss of calcite between the production by calcifier organisms in the euphotic zone and the estimated calcite flux below the lysocline [81]. This calcite loss could be attributed to biological activities and more specifically the dissolution mediated by zooplankton grazing or transport. Several studies have even shown a loss of calcite after zooplankton gut passage, a striking feature of the sedimentary record that relies on the observation of well-preserved coccoliths within zooplankton fecal pellets [79,82–85]. However, numerical models using a timeframe and pH inside copepod guts suggest a moderate calcite dissolution inside the gut [86]. Langer et al. [87] showed that calcite dissolution during copepod gut passage was below 8% of the weight of the coccoliths of *Calcidiscus leptoporus* inside fecal pellets, but these coccoliths were intact and showed no evidence of any dissolution [87]. In addition, Antia et al. [88] successfully observed that coccolith dissolution/fragmentation occurs inside zooplankton guts and microzooplankton vacuoles. During the first experiment, we observed a decoupling between ingestion rates (both pigment ingestion rates and equivalent volume ingestion rates) and fecal pellet egestion (Figure 3). Taking all the experiments collectively, this fact was also noticed in Experiment 5, with a high cell concentration of *E. huxleyi*. In addition, despite high measured ingestion rates, few fecal pellets were produced (Figures 5, 7 and 8). This decoupling between ingestion and egestion could be the result of a modulation of the residence time in the gut. Hence, fecal pellet size seems to depend on the ingestion rate index and prey quality (Figures 1, 6 and 8). Indeed, the fecal pellet size variation could depend on gut passage time as well [89]. By considering all these

points, both coccolithophore biovolume and relative calcite content may modulate coccolith dissolution due to gut passage variations.

### 4.4. Consequences for Vertical Particle Flux in the Pelagic Realm

In this study, we observed a loss of fecal pellet production with high prey concentrations, despite high ingestion rates. The number of egested particles (fecal pellets) seems to be dependant, not only on the food quantity, but also on the quality of the ingested food. The size of egested particles could be increased by both the number of ingested particles and their quality (inclusion of calcite, silica frustules, etc.). Prey/predator size ratio and relative carbon content [90] suggest that these environmental food conditions may provide predictable constraints to copepod biogeography size distribution in the ocean [91]. This therefore suggest that size and primary producer stoichiometry could influence oceanic carbon flux patterns through fecal pellet egestion by copepods. This may result in a decrease of carbon passive flux due to fecal pellets sinking in the water column. In addition, if we consider that fecal pellets follow Stoke's law of sedimentation (as suggested by Komar et al. [92]), the ballast effect of calcified coccoliths inside fecal pellets should foster the sedimentation rate much more than changes in the size of the pellets [93]. Hence, modification of fecal pellet egestion patterns in addition to ballast effect of calcite could be an important process driving the particle flux in the water column.

## 5. Conclusions

In this study, we demonstrated that copepod ingestion rates based on the volume equivalent of cells is better scaled to total prey volume concentration ($mm^3 \, L^{-1}$). The Chl *a*/biovolume calibration developed in this study highlights the importance of considering the Chl *a* level inside the gut fluorescent content, regarding food types ingested by wild copepods, such as non-chlorophyllian preys (e.g., microzooplankton, heterotrophic flagelles, nauplii, etc.). Our results highlight an exponential increase of ingestion rates according to food availability (*Type III functional response*), which is in contrast to the optimal Ivlev model (*Type II functional response*) corresponding to optimal food foraging. This parametric pattern supports the role of food quality in the feeding behaviour of copepods, such as coccolithophore defence structures (calcified coccospheres). We demonstrated this aspect by showing the relationship between calcite content (PIC/POC ratio) and the ingestion rate index. Finally, we observed a decoupling between ingestion rates and fecal pellet egestion, which may be the consequence of an "obstruction" effect of calcite inside the copepod's gut. This "obstruction" may be the result of varying gut passage times—modulating the intensity of calcite dissolution. These results suggest that both prey allometry and stoichiometry need to be considered with copepod feeding dynamics, specifically regarding fecal pellet production, and the sedimentary flux, which is an important component of the biological carbon pump.

**Author Contributions:** Contributed to conception and design: J.T. and A.D. Contributed to acquisition of data: J.T., A.D., A.P. and G.D. Contributed to the copepod rearing maintenance: J.T., A.D. and A.P. Contributed to analysis and interpretation of data: J.T., A.D. and M.H. Drafted and/or revised the article: J.T., A.D. and M.H. Approved the submitted version for publication: J.T., A.D., A.P., G.D., V.C., L.B. and M.H. All authors have read and agreed to the published version of the manuscript.

**Funding:** This work was supported by the ANR CARCLIM (https://anr.fr/Projet-ANR-17-CE01-0004). Jordan Toullec's postdoctorate was equally funded by ANR CARCLIM and by Université Littoral Côte d'Opale. This work was also supported by the CPER MARCO and the SFR Campus de la Mer.

**Institutional Review Board Statement:** Not applicable.

**Informed Consent Statement:** Not applicable.

**Data Availability Statement:** All data are displayed within the manuscript or may be obtained directly from the authors.

**Acknowledgments:** We would like to thank the Sepia II crew for their help at sea, during zooplankton collection, and more specifically for the multiple WPII plankton hauls during *Phaeocystis globosa* sticky bloom event. We are also grateful to Michel Laréal for his help and modifications made to the rolling table.

**Conflicts of Interest:** The authors declare no conflict of interest.

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
