# Peer review of "Copepod Feeding Responses to Changes in Coccolithophore Size and Carbon Content"

_jmse, doi:10.3390/jmse10121807_

Round 1

Reviewer 1 Report

I think that the ms is an excellent study to understand copepods' functional feeding.

In materials and methods, the types of the functional response for zooplankton food availability need to be explained.

References should be re-edited according to JMSE style.

Author Response

Thank you for the review and comments.

The different copepod functional responses cited (e.g: omnivorous feeding-current feeders, ambush feeders or cruise feeders...) are not relative to food availability. These are functional traits (described in Benedetti et al., 2016; Lombard et al, 2013 and references in here), and are more relative to species niche. These publications also refers to more descriptive literature about specific copepods mobility/3D dynamic. Basically we have to consider active feeding behavior for the two copepods species used during the experiments (T. longicornis and A. clausi). However T. longicornis have also the possibility to track particles (sinking particles) and so adapt its feeding strategy. 

The references has been re-edited according to JMSE style, thanks for the notice.

Reviewer 2 Report

This manuscript reports an ecological study of the impacts of phytoplankton stoichiometry and size on copepod feeding dynamics. What sets this study apart from many other ecological studies is its focus on the process leading to the formation of fecal pellets. The conclusion is that both prey size and stoichiometry have impacts on copepod feeding dynamics, specifically the process leading to the formation of fecal pellets. In my opinion, this work is interesting and very significative. However, the manuscript needs careful editing as shown below. Therefore, I recommend to consider major revision.

The specific opinions are as follows.

Major questions

Question 1 Cell size (colony size) of phytoplankton species play important roles in determining the function and stability of plankton ecosystems. In this study, I think you should give more analysis referring the mechanisms of cell size or colony size of coccolithophores on grazing rate of copepods, and on phytoplankton-zooplankton dynamics in both the parts of introduction and discussion. You could refer to these articles:

Pan Y, Zhang Y.S, Sun S.C. 2014. Phytoplankton-zooplankton dynamics vary with nutrients: a microcosm study with Coleofasciculus chthonoplastes and Moina micrura. Journal of Plankton Research, 36(5): 1323 -1332. 

Wan LL, Long YY, Hui J, Zhang H, Hou ZY, Tan JX, Pan Y, Sun SC. 2020. Effect of norfloxacin on algaecladoceran grazerlarval damselfly food chains: algal morphology-mediated trophic cascades. Chemosphere, 2020, 127166.                             

Pan Y, Dong JY, Wan LL, Sun SC, MacIsaac HJ, Drouillard KJ, Chang XX. 2019. Norfloxacin pollution alters species composition and stability of plankton communities. Journal of Hazardous Materials. 385 (2020) 121625. 

Question 2 The writing needs work before it is up to publish standards. The logic confusion and the misuse of important information create confusion about what the authors want to say in a number of places.

About the title

n  “Copepod functional responses” is not clear, what kinds of function do you refer here. 

About the Abstract

n  Line 10-11, “Stoichiometry of phytoplankton and size could result from both phenology and environmental change”, language problem, size of what? Do you mean phytoplankton stoichiometry and size?

n  For the first two sentences, you should declare the innovation of this research. 

n  Line 18-19, I don’t understand the purpose and significance of this result. 

n  Line 19-21, how coccolithophore biovolume and particulate inorganic/organic carbon ratios affect food foraging for copepods? You should describe the result more clarity. 

n  Line 21-22 also, what do you mean a decoupling between ingestion rates and fecal pellet egestion.

About the Introduction

n  64-66. “provide an anti-grazing function by copepod”, not clear here. Do you mean that act as a morphological strategy to defend against the copepods. 

n  Line 70-74. It would be better if you could give some examples to explain how phytoplankton size or stoichiometry affect zooplankton feeding rate, and thus on phytoplankton-zooplankton dynamics, system stability or even trophic cascade. Please see the references below. 

1) Pan Y, Zhang Y.S, Sun S.C. 2014. Phytoplankton-zooplankton dynamics vary with nutrients: a microcosm study with Coleofasciculus chthonoplastes and Moina micrura. Journal of Plankton Research, 36(5): 1323 -1332.

2) Wan LL, Long YY, Hui J, Zhang H, Hou ZY, Tan JX, Pan Y, Sun SC. 2020. Effect of norfloxacin on algaecladoceran grazerlarval damselfly food chains: algal morphology-mediated trophic cascades. Chemosphere, 2020, 127166.

3) Pan Y, Dong JY, Wan LL, Sun SC, MacIsaac HJ, Drouillard KJ, Chang XX. 2019. Norfloxacin pollution alters species composition and stability of plankton communities. Journal of Hazardous Materials. 385 (2020) 121625. 

n  In part of introduction, although you introduce the diatom silicification and biomineral shells made of calcium carbonate, your manuscript did not contain the details about the relationships between stoichiometry of phytoplankton species and the capacity of these species in defending themselves against grazers.

n  Besides, throughout this part, you did not give any information associated with the process of fecal pellet egestion.

About the Method

n  For table 1, what do you mean exp. 1 to exp.5. Do you mean experiment 1 to 5? What is your experimental design? 

n  I think information about your experimental design should be described before all experimental procedures and set as a separate chapter to provide convenience for reads. 

n  Line 204-206. Is this method of determination reliable? How to eliminate the interference of other factors, such as the residue of the molt and death copepod, or even algal cells and microorganisms. 

About the Results

n  Line 279-280 In part of result, we usually do not contain references. 

n  Line 282-284 in part of result, we also do not contain about information of experiment design. Please remove this information to part of method. 

n  Line 399 and 416 In part of result, we usually do not contain references.

About the discussion

n  Line 431 and 432 This sentence is very difficult to understand.

n  Line 432 and 434, this relationship also exists in freshwater ecosystems.

n  Line 431-438, I think it is better to introduce the different between this study and previous works in order to highlight the innovation of this article.

I hope that these comments help.

With regards

Author Response

Comments and Suggestions for Authors

This manuscript reports an ecological study of the impacts of phytoplankton stoichiometry and size on copepod feeding dynamics. What sets this study apart from many other ecological studies is its focus on the process leading to the formation of fecal pellets. The conclusion is that both prey size and stoichiometry have impacts on copepod feeding dynamics, specifically the process leading to the formation of fecal pellets. In my opinion, this work is interesting and very significative. However, the manuscript needs careful editing as shown below. Therefore, I recommend to consider major revision.

The specific opinions are as follows.                                                                    

Author response We want to thanks the Reviewer 2 for the comments and suggestions, we are convinced that these revisions have considerably improve the manuscript. 

Major questions

Question 1 Cell size (colony size) of phytoplankton species play important roles in determining the function and stability of plankton ecosystems. In this study, I think you should give more analysis referring the mechanisms of cell size or colony size of coccolithophores on grazing rate of copepods, and on phytoplankton-zooplankton dynamics in both the parts of introduction and discussion. You could refer to these articles:

Pan Y, Zhang Y.S, Sun S.C. 2014. Phytoplankton-zooplankton dynamics vary with nutrients: a microcosm study with Coleofasciculus chthonoplastes and Moina micrura. Journal of Plankton Research, 36(5): 1323 -1332. 

Wan LL, Long YY, Hui J, Zhang H, Hou ZY, Tan JX, Pan Y, Sun SC. 2020. Effect of norfloxacin on algae–cladoceran grazer–larval damselfly food chains: algal morphology-mediated trophic cascades. Chemosphere, 2020, 127166.                             

Pan Y, Dong JY, Wan LL, Sun SC, MacIsaac HJ, Drouillard KJ, Chang XX. 2019. Norfloxacin pollution alters species composition and stability of plankton communities. Journal of Hazardous Materials. 385 (2020) 121625. 

Author response Question 1: Thank you for the comment. do you mean cell concentration when you say ‘colony size’? Or do you consider colony such as cell chain of EPS aggregated cells (E.G: Phaeocystis spp.). There is some relevant works about diatom chain formation or EPS aggregated cells and copepod grazing. However, in our study, we considered coccolithophores which are not forming colonies. Moreover, we have considered cell concentration as a major determinant of copepod grazing regarding the optimal food foraging theory (basically, the more food is available, the more the copepods eat). In this context, we wanted to examine the implication of calcified structure onto grazing, independently of the concentration of food. The suggested reference about nutrients implication into Phytoplankton/Zooplankton dynamics is interesting and has been added to the manuscript (line 68).

Question 2 The writing needs work before it is up to publish standards. The logic confusion and the misuse of important information create confusion about what the authors want to say in a number of places.

Author response Question 2: We have taken into consideration every reviewers’ comment/suggestion, we are convinced that the manuscript quality is better now.

About the title

n  “Copepod functional responses” is not clear, what kinds of function do you refer here. 

Author response You are right, there is a word missing to make it clearer: We change “functional” by “feeding” which is clearer about what we’ve done:  Copepod feeding responses to changes in coccolithophore size and carbon content

About the Abstract

n  Line 10-11, “Stoichiometry of phytoplankton and size could result from both phenology and environmental change”, language problem, size of what? Do you mean phytoplankton stoichiometry and size?

Author response Yes, we meant phytoplankton cell size, we added “cell size”

n  For the first two sentences, you should declare the innovation of this research. 

Author response I think the third sentence expresses the innovation of this research “We performed incubations with copepods and coccolithophores including different prey sizes and particulate carbon contents “The Abstract have been reorganized in order to highlight the innovative aspect of our study.

n  Line 18-19, I don’t understand the purpose and significance of this result. 

Author response In our study we considered the Chl a/biovolume ratio of phytoplankton, within copepods gut fluorescence, and we proposed a correction of copepods ingestion rate based on guts fluorescence (considering Chl a/biovolume ratio). A precision has been added by modifying the abstract

n  Line 19-21, how coccolithophore biovolume and particulate inorganic/organic carbon ratios affect food foraging for copepods? You should describe the result more clarity. 

Author response Coccolithophore biovolume and particulate inorganic/organic carbon ratios limit food foraging by copepods. We observe a decrease of food ingestion when PIC/POC. We slightly modify the sentence: “both coccolithophore biovolume and particulate inorganic/organic carbon ratios affect the food foraging for copepods.”

n  Line 21-22 also, what do you mean a decoupling between ingestion rates and fecal pellet egestion.

Author response We observe a nonlinear variation between ingested rate and fecal pellet egestion rate, basically despite copepods have high ingestion rate, we don’t always observe the higher fecal pellet egestion rate. We modify the sentence: “Last, we observed a non linear relation between ingestion rates and fecal pellet egestion”.

About the Introduction

n  64-66. “provide an anti-grazing function by copepod”, not clear here. Do you mean that act as a morphological strategy to defend against the copepods. 

Author response n  64-66: We meant that coccospheres structure are effectively a morphological strategy against grazing (such as shell). We modified the sentence by “coccosphere could also be considered as an anti-grazing function by copepod”.

n  Line 70-74. It would be better if you could give some examples to explain how phytoplankton size or stoichiometry affect zooplankton feeding rate, and thus on phytoplankton-zooplankton dynamics, system stability or even trophic cascade. Please see the references below. 

1) Pan Y, Zhang Y.S, Sun S.C. 2014. Phytoplankton-zooplankton dynamics vary with nutrients: a microcosm study with Coleofasciculus chthonoplastes and Moina micrura. Journal of Plankton Research, 36(5): 1323 -1332.

2) Wan LL, Long YY, Hui J, Zhang H, Hou ZY, Tan JX, Pan Y, Sun SC. 2020. Effect of norfloxacin on algae–cladoceran grazer–larval damselfly food chains: algal morphology-mediated trophic cascades. Chemosphere, 2020, 127166.

3) Pan Y, Dong JY, Wan LL, Sun SC, MacIsaac HJ, Drouillard KJ, Chang XX. 2019. Norfloxacin pollution alters species composition and stability of plankton communities. Journal of Hazardous Materials. 385 (2020) 121625. 

Author response n  70-74: Thanks for the comment. Some studes explain the specific copepod responses toward phytoplankton size or stoichiometry, which led to trophic cascades. They have been cited in the introduction:

1) Nejstgaard, J.C.; Gismervik, I.; Solberg, P.T. Feeding and Reproduction by Calanus Finmarchicus, and Microzooplankton Grazing during Mesocosm Blooms of Diatoms and the Coccolithophore Emiliania Huxleyi. Marine Ecology Progress Series 1997, 147, 197–217. (line XX)

2) Stibor, H.; Vadstein, O.; Diehl, S.; Gelzleichter, A.; Hansen, T.; Hantzsche, F.; Katechakis, A.; Lippert, B.; Løseth, K.; Peters, C.; et al. Copepods Act as a Switch between Alternative Trophic Cascades in Marine Pelagic Food Webs. Ecology Letters 2004, 7, 321–328, doi:10.1111/j.1461-0248.2004.00580.x. (line XX)

3) Moriceau, B.; Iversen, M.H.; Gallinari, M.; Evertsen, A.-J.O.; Le Goff, M.; Beker, B.; Boutorh, J.; Corvaisier, R.; Coffineau, N.; Donval, A.; et al. Copepods Boost the Production but Reduce the Carbon Export Efficiency by Diatoms. Frontiers in Marine Science 2018, 5, doi:10.3389/fmars.2018.00082. (line XX)

However, we added the reference about cladoceran/ cyanobacteria interaction (Pan et al., 2014), which is interesting to extent the cited references to other zooplankton group inside “trophic cascade” dynamic and biogeochemistry.

Nevertheless, marine copepod functional traits, especially the feeding behavior are now well described in the literature. other zooplankton/phytoplankton interactions, such as cladocerans and other phytoplankton groups (e.g Cyanobacteria) are not necessarily comparable with specific copepod trophodynamic. (line 68).

n  In part of introduction, although you introduce the diatom silicification and biomineral shells made of calcium carbonate, your manuscript did not contain the details about the relationships between stoichiometry of phytoplankton species and the capacity of these species in defending themselves against grazers.

Author response We manly used PIC/POC ratio as proxy of coccolithophore calcification, and you are right, it is not really a stoichiometric ratio. Indeed, we did not discuss about C/N or N/P ratio as a driver of copepod ingestion, which are more related to cell quality health, regarding lipid/protein content (these process as already been well described in the literature). In our case we fill in the literature gap concerning the relationship between calcification (PIC/POC) and copepod grazing (as it has already been done with diatom frustules).

n  Besides, throughout this part, you did not give any information associated with the process of fecal pellet egestion.

Author response You are right, we neglected to mention the process responsible to pellet production. We added this sentence: “Moreover food type and availability affect fecal pellet production rates, pellet volumes, and sinking rate, regarding compactness and mineral ballasting (Besiktepe et Dam, 2002; Butler et Dam, 1994)”. Lines 74-76

About the Method

n  For table 1, what do you mean exp. 1 to exp.5. Do you mean experiment 1 to 5? What is your experimental design? 

Author response We did a set of 5 experiments (exp. 1 to 5), which correspond to 9 incubations with copepod during exp. 1 and 6 incubation with copepod for exp 2 to 5. On the whole we performed 33 incubations with copepods. To be less confusing we change “exp x” to “x” in table 1.

n  I think information about your experimental design should be described before all experimental procedures and set as a separate chapter to provide convenience for reads. 

Author response Thank you for the suggestion. I split the 2.3 Experimental incubation paragraph into different sub-paragraph in order to clarify the experimental design.

n  Line 204-206. Is this method of determination reliable? How to eliminate the interference of other factors, such as the residue of the molt and death copepod, or even algal cells and microorganisms. 

Author response After each incubation, all the copepods were flash feezed (in order to avoid fecal material egestion and stop pigments degradation) and then kept at -20°C until the fluorescence measure. We didn’t have a lot of dead recovered copepods after the incubation (less than 20%), and dead copepod recover were also used for residual pigment measures (you are right, it should necessarily include a bias which cannot be avoided even when copepods are rinsed in filtered sea water). Concerning the residual cell stuck on the molt (which are fluorescent as well), it is negligible regarding the gut content (which were very dense).

We also measured “blank” copepod residual fluorescence (copepod with empty guts), even with a bulk of 30 empty copepod per acetone extraction, we did not have any fluorescence signal indicating that cells stucked in the molt could be neglected in this study. This method is thus relatively reliable.

About the Results

n  Line 279-280 In part of result, we usually do not contain references. 

Author response We removed the sentence with reference, and replace it into the discussion.

n  Line 282-284 in part of result, we also do not contain about information of experiment design. Please remove this information to part of method. 

Author response We removed this sentence, and replace it to discussion part (line 384-388), because this statement refers to an experimental limit of our setup (regarding copepod feeding saturation).

n  Line 399 and 416 In part of result, we usually do not contain references.

Author response We are convinced that the cited reference about the different type of functional responses (Jeschke et al., 2004) should stay in this result part, as we use the Type I,II and III model to described our results.

(Jeschke, J.M.; Kopp, M.; Tollrian, R. Consumer-Food Systems: Why Type I Functional Responses Are Exclusive to Filter Feeders. Biological Reviews 2004, 79, 337–349, doi:10.1017/S1464793103006286.)

About the discussion

n  Line 431 and 432 This sentence is very difficult to understand.

Author response You are right. We change this sentence to: “Thus, we suggest that size and the stoichiometric of primary producers could control oceanic carbon flux patterns thank to fecal pellet egestion.”

n  Line 432 and 434, this relationship also exists in freshwater ecosystems.

n  Line 431-438, I think it is better to introduce the different between this study and previous works in order to highlight the innovation of this article.

Author response We switched the order of the paragraph:

“In this study, we observed a loss of fecal pellet production with high prey concentrations, despite high ingestion rates. The number of egested particle (fecal pellets) seems to be dependent, not only on the food quantity, but also on the quality of the ingested food. The size of egested particles could be increased by both the number of ingested particles and their quality (inclusion of calcite, silica frustules…). Prey/predator size ratio and relative carbon content [91] suggest that these environmental food conditions may provide predictable constraints to copepod biogeography size distribution in the ocean [92]. Thus, we suggest that size and the stoichiometric of primary producers could control oceanic carbon flux patterns thank to fecal pellet egestion”

I hope that these comments help.

Author response Thank you again for your comments and revisions, we are convinced that the manuscript quality has been considerably improved.

With regards
